# Childbearing Age Women Characteristics in Latin America. Building Evidence Bases for Early Prevention. Results from the ELANS Study

**DOI:** 10.3390/nu13010045

**Published:** 2020-12-25

**Authors:** Marianella Herrera-Cuenca, Agatha Nogueira Previdelli, Berthold Koletzko, Pablo Hernandez, Maritza Landaeta-Jimenez, Yaritza Sifontes, Georgina Gómez, Irina Kovalskys, Martha Cecilia Yépez García, Rossina Pareja, Lilia Yadira Cortés, Attilio Rigotti, Mauro Fisberg

**Affiliations:** 1Center for Development Studies, Central University of Venezuela (CENDES-UCV), Caracas 1050, Venezuela; 2Bengoa Foundation for Food and Nutrition, Caracas 1071, Venezuela; mlandaetajimenez@gmail.com (M.L.-J.); yaritza.sifontesv@gmail.com (Y.S.); 3Faculty of Biological and Health Sciences, University São Judas Tadeu, São Paulo 03166-000, Brazil; agatha.usp@gmail.com; 4Departamento Paediatrics, Division Metabolic and Nutritional Medicine, Dr. von Hauner Children’s Hospital, LMU University of Munich, D-80377 Munich, Germany; berthold.koletzko@med.uni-muenchen.de; 5School of Nutrition and Dietetics, Faculty of Medicine, Central University of Venezuela, Caracas 1041-A, Venezuela; doctuscumliber@gmail.com; 6Department of Biochemistry, School of Medicine, University of Costa Rica, San José 11501-2060, Costa Rica; georgina.gomez@ucr.ac.cr; 7Faculty of Medicine, The Pontifical Catholic University of Argentina, Buenos Aires C1107AAZ, Argentina; ikovalskys@gmail.com; 8College of Health Sciences, San Francisco de Quito University, Quito 17-1200-841, Ecuador; myepez@usfq.edu.ec; 9Nutrition Research Institute, La Molina, Lima 15026, Peru; rpareja@iin.sld.pe; 10Department of Nutrition and Biochemistry, Pontificia Universidad Javeriana, Bogotá 110111, Colombia; ycortes@javeriana.edu.co; 11Department of Nutrition, Diabetes and Metabolism, School of Medicine, Pontificia Universidad Católica, Santiago 833-0024, Chile; arigotti@med.puc.cl; 12Instituto Pensi, José Egydio Setubal Foundation, Sabará Children’s Hospital, São Paulo 01239-040, Brazil; 13Department of Pediatrics, Paulista School of Medicine, Federal University of São Paulo, São Paulo 04023-062, Brazil

**Keywords:** childbearing age women, Latin America, nutritional status, food consumption, physical activity

## Abstract

Latin American (LA) women have been exposed to demographic and epidemiologic changes that have transformed their lifestyle, with increasing sedentary and unhealthy eating behaviors. We aimed to identify characteristics of LA women to inform public policies that would benefit these women and their future children. The Latin American Study of Nutrition and Health (ELANS) is a multicenter cross-sectional study of representative samples in eight Latin American countries (*n* = 9218) with a standardized protocol to investigate dietary intake, anthropometric variables, physical activity, and socioeconomic characteristics. Here we included the subsample of all 3254 women of childbearing age (15 to <45 years). The majority of ELANS women had a low socioeconomic status (53.5%), had a basic education level (56.4%), had a mostly sedentary lifestyle (61.1%), and were overweight or obese (58.7%). According to the logistic multiple regression model, living in Peru and Ecuador predicts twice the risk of being obese, and an increased neck circumference is associated with a 12-fold increased obesity risk. An increased obesity risk was also predicted by age <19 years (Relative Risk (RR) 19.8) and adequate consumption of vitamin D (RR 2.12) and iron (RR 1.3). In conclusion, the identification of these risk predictors of obesity among Latin American women may facilitate targeted prevention strategies focusing on high-risk groups to promote the long-term health of women and their children.

## 1. Introduction

The foundations for a healthy life are laid early, from pre-conception through pregnancy and early childhood. Maternal nutritional status at the time of oocyte implantation is a key determinant factor of embryonic and fetal growth [1,2]. Previous studies indicated that nutritional interventions prior to and during pregnancy can positively influence a newborn’s birthweight and well-being [3,4,5], making early dietary and lifestyle interventions a key strategy for achieving optimal population health. Obese women and their future children also benefit from interventions to achieve optimal weight before pregnancy, since obese childbearing-age women might have impaired fertility and increased risk for excessive weight gain during pregnancy, maternal complications, Cesarean delivery and fetal macrosomia, and ultimately higher mortality [6,7].

Obesity and undernutrition are key relevant factors for women of reproductive age as determinants for higher risks of chronic diseases later in life in the woman and in her offspring [8]. Food is an environmental exposure contributing to epigenetic modifications [9]. Global trends in dietary quality and micronutrient supplies reports 2.1 billion people are overweight or experiencing obesity, 2 billion have micronutrient deficiencies, and 795 million people live in hunger. All of the above include at some point women of reproductive age [10]. A major public health concern is the marked rise of obesity in low- and middle-income countries (LMIC). As an example, 20 million women were obese in India in 2014, in contrast to only 0.8 million in 1975 [11]. In addition, in WHO geographic locations of LMIC the estimated prevalence of obesity in women of fertile age varied according to region: from 39.5% in Egypt (the eastern Mediterranean) to 1% in Ethiopia (Africa) and 7.8% in Haiti (the Americas) [11]. In contrast, the median prevalence of underweight women aged 20–49 years from 36 developing countries was 5.9% and 9.3% for urban and rural areas, respectively [12]. Additionally, in 2016, 32.8% of global women of reproductive age (15–49 years) were affected by iron deficiency anemia [13].

Latin American populations have been exposed to rapid demographic, epidemiologic, and nutritional changes. These include a transformation towards a more sedentary lifestyle, an increased participation in the workforce, and moving towards urban life, yet many must deal with living in poverty and social inequalities that contribute to increased susceptibility to diseases [14].

Epidemiologic studies during the last decades have identified several markers associated with lifestyle factors related to future health risks, among those low socioeconomic status, poor education, limited diet diversity, short stature, low and high weight, and body mass index [1,15,16]. Diet is an important factor in modulating epigenetic markers and appears to link gene regulation to social living conditions [9]. In addition to diet, lack of physical activity, sleep deprivation, or living in poor conditions might influence epigenetic markers and related health outcomes. Thus, many environmental signals impact the development of disease. The consequences on the health of women that may become mothers can impact the health of future generations, and might be an opportunity for applying preventive strategies [17].

The aim of this study was to identify lifestyle and dietary characteristics of women of childbearing age through sociodemographic characteristics; dietary intake of energy, proteins, and micronutrients; physical activity; and anthropometric characteristics.

## 2. Materials and Methods

The Latin American Study in Nutrition and Health/*Estudio Latinoamericano de Nutrición y Salud* (ELANS) is a household-based, multicentric cross-sectional survey aimed at describing the nutritional and anthropometric status of Latin American populations, with an assessment of food and nutrient intake and physical activity levels of representative samples from urban populations in eight countries (Argentina, Brazil, Chile, Colombia, Costa Rica, Ecuador, Peru, and Venezuela). The fieldwork for the ELANS study was conducted from 2014 to 2015 [18].

### 2.1. Sample

The ELANS study enrolled 9218 adolescents and adults aged 15 to 65 years, including 3259 women aged between 15 and 45 years. Sampling was random complex multistage, stratified by geographical location, gender, age, and social economic level (SEL) only for the urban population [18]. For this analysis we selected all participating women of childbearing age from 15 to <45 years [19]. Subjects were stratified into subgroups by age from 15–<19 years (adolescents), 19–<35 years (young women), and 35–<45 years (older women).

All subjects gave their informed consent for inclusion before they participated in the study. The study was conducted in accordance with the Declaration of Helsinki, and the protocol was approved by the Western Institutional Review Board (#20140605). Ethics review boards of participating institutions approved this study, and it was registered in Clinical Trials (#NCT02226627).

### 2.2. Sociodemographic

Sociodemographic variables were collected through a questionnaire. Information on age (years), gender, years of education, race/ethnicity, and marital status was obtained. Socioeconomic level was categorized as low, middle, and high [18].

### 2.3. Dietary Assessment

The dietary assessment was conducted during 2 separate household visits on non-consecutive days, with an interval of up to 8 days between them. During both visits, a 24 h dietary recall (24-H) was conducted using the Multiple Pass Method [20] to assess all foods and beverages consumed over the prior day. The 24-H recalls included both weekdays and weekend days. The household’s measures obtained in the 24-H were converted to grams and milliliters by trained nutritionists.

Energy, macronutrient, and micronutrient intakes were calculated from the food and beverage intake with the Nutrition Data System for Research (version 2013 software (NDS-R), University of Minnesota, MN). A food-matching standardized procedure was conducted by professional nutritionists in each country. The complete food standardization procedure has been described in detail [21].

Regional foods, recipes, and commercial foods not available in the NDS-R database were broken down into ingredients and entered in the software as user recipes. These user recipes were created from the available NDS-R database and documented in food-matching control sheets. They were obtained from national publications, recipe books, and culinary websites of each country and checked against actual data from 24-H. When regional foods did not have an exact equivalent or similar food available in the NDS-R database, one or more foods combined were inserted as a recipe, as the software does not accept “new foods.” Local teams were responsible for creating a recipe that represented the same nutritional value as the original version. All food and beverages reported in both 24-H, considering all countries, were coded by the NDS-R software.

#### 2.3.1. Dietary Intake

Two 24-H recalls were used to estimate habitual food consumption and to evaluate intra-individual variability in nutrient intake. The web-based statistical modeling technique Multiple Source Method (MSM) (https://msm.dife.de/tps/en), proposed by the European Prospective Investigation into Cancer and Nutrition (EPIC), was applied to estimate the habitual intake of energy and macronutrients. This method was chosen because of its capability to improve estimates of usual dietary intake of energy, nutrients, foods, and food groups by considering within-person variance in intake, thereby improving the usual intake distribution for the population [20]. Due to differences in eating habits among the Latin American population, an estimation of regular intake was conducted separately for each country. The relative contribution of each macronutrient to total energy intake was subsequently calculated [18].

#### 2.3.2. Micronutrient Intake Analysis

Based on the 24-H recalls, the intake of iron, calcium, and vitamins A, C, and D was analyzed and compared to the US Institute of Medicine estimated required allowances (ERA) to assess adequacy [22].

### 2.4. Anthropometry

Categorization of body mass index (BMI) values in adolescents (15 to <19 years) was based on cut-offs based on the sex-specific WHO BMI-for-age growth charts [23], with underweight defined as (BMI < –2SD), normal weight (–2SD ≥ BMI ≤ 1 SD), overweight (1SD ≥ BMI ≤ 2 SD), and obese (BMI for age > 2SD). For adult women (19 years and older), BMI was categorized as underweight (<18.5 kg/m²), normal weight (18.5–24.9 kg/m²), overweight (25–29.9 kg/m²), and obese (≥30.0 kg/m²) [24]. The study population was categorized as having excess weight (overweight and obese) or no excess weight (underweight or normal weight). Waist circumference was assessed based on the validated cutoff points of the Latin American Study Group of the Metabolic Syndrome (GLESMO) for Latin American populations [25]. Neck circumference (NC) was categorized as abnormal if the circumference was >34.5 cm for boys and >31.25 for girls [26], whereas for adults the cutoff points for abnormal were >39 cm for men and >35 cm for women. Women’s height was categorized as short (< or equal to 155 cm) or normal (>155 cm), taking into account that the population studied in this project was urban and included all SELs across the countries [27].

### 2.5. Physical Activity Assessment

#### Self-Reported Activities

Self-reported physical activity was assessed using the International Physical Activity Questionnaire (IPAQ)—long version, a validated self-report measurement tool for physical activity in Latin America [28]. The Mexican (Spanish language) version of IPAQ [29] was adapted for all countries of the ELANS, using culturally appropriate wording and examples. Only the leisure time and transport physical activity (LTPA and TPA) sections were included, due to the greater importance of these domains in public health and poor validity of occupational and home-based physical activity IPAQ sections in Latin American urban settings. These sections are the most relevant for categorizing population levels of physical activity and for guiding public health policies and programs [28].

A domain-specific activity score was calculated separately for each domain of physical activity (transportation and leisure time). Total times engaged in walking, moderate physical activity, and vigorous physical activity, all expressed in min/week, were scored using established methods posted on the IPAQ website (www.ipaq.ki.se).

Additionally, information not included as part of the summary score of physical activity, such as sedentary activities (reading, television viewing, and sitting at a desk), were analyzed.

### 2.6. Statistics

Descriptive statistical analysis was carried out in the present study. Categorical variables were expressed as absolute numbers of cases and percentage values. Central tendency and dispersion were summarized using means, standard deviations, and percentiles (P25, median, and P75).

Multiple logistic regression models were performed to evaluate associations between excess weight and covariates while controlling for misreporting of energy intake [30] and socioeconomic level. Independent variables with *p*-values ≤ 0.20 in univariate analysis were selected for multiple regression analyses and included in the regression model by the stepwise forward procedure. Variables that remained significant in the multiple logistic regression model (*p* < 0.05) were maintained. A probability value of 0.05 was considered statistically significant. All statistical analyses were performed using STATA 12.0 (Stata Corporation, College Station, TX, USA).

## 3. Results

We analyzed data from 3259 women of childbearing age, defined as aged between 15 and 44 years. Nutritional status, socioeconomic characteristics, educational level, and marital characteristics are shown in Table 1. The majority of the women were within the low socioeconomic level (53.5%), had a basic educational level (56.4%), and were married or lived with a partner (49.3%). Venezuela and Colombia had the highest values of women within the low socioeconomic level, whereas Peru, Costa Rica, and Ecuador showed a prevalence >10% of high socioeconomical level. The educational level of Venezuela was considerably higher than in other countries. Superior educational level was reported by 20.7% of Venezuelan women, nearly twice the mean proportion in the total ELANS group.

In general, older women presented higher values in all the anthropometric measures than younger women (Table 2). The majority of women were overweight or obese (58.7%), especially in Costa Rica (60.7%) and Chile (60.0%), which also had the highest proportions of increased waist circumference (65.9% and 59.7%) and neck circumference (38.2% and 47.4%), respectively (Table 3). Venezuela and Brazil had the highest proportions of underweight women (5.3% and 4.8%, respectively). The greatest prevalence for stunting (<155 cm) was identified in Peru (60.30%) and Ecuador (54.55%) (mean: 153.30 ± 5.54 cm and 154.81 ± 6.53 cm, respectively).

The majority of the studied women in LA reported a predominantly sedentary lifestyle (61.12%), particularly in Venezuela (73.49%) and Brazil (70.66%); the exception was Ecuador, which represented the country with the highest proportion of physically active women (66.07%). With increasing age, energy intake tended to decrease when the proportion of protein increased (Table 4). Proteins provided an average of 15–16% of energy intake, fats 29–30%, and carbohydrates 53–54%. Most of the countries showed mean protein intake matching recommended intake. Argentina, Colombia, Perú, Chile, and Venezuela had more than 20% of their female population with a fat intake higher than the recommended intake range, however, for Peru it should be highlighted that it was the only country with a mean fat intake below recommendations, making it clear that mean values should be cautiously analyzed, as some values might be at extremes, thus making the average not fully demonstrative of the majority (Figure 1). Peru also had the highest proportion of the population with a carbohydrate intake above the recommended range of intake (28.9%).

Calculated micronutrient intakes are shown in Table 5 by age groups. Figure 2 shows the percentages of individuals with adequate calcium and iron intake. Most women had an inadequate calcium intake, particularly in Peru (97.74%), Costa Rica (95.88%), Brazil (95.16%), and Chile (93.23%). The highest proportion of inadequate iron intake was found in Brazil (40.5% of the population).

Table 6 presents associations between the risk of excess weight and different predictors. Age group, neck circumference, country of residence, and vitamin D and iron intake were relevant predictors in the final regression model. Women aged 19 to 24.9 years had a 9.87-fold higher risk for excess weight and those aged 35–44.9 years had a 19.80-fold higher risk for excess weight than female adolescents. An increased neck circumference predicted a 12-fold increase of obesity risk.

The country of residence was also important: Living in Ecuador or Peru predicted twice the risk for excess weight when compared to Argentina. Additionally, adequate intakes of vitamin D and iron were associated with a 2.12- and 1.34-fold increased risk for excess weight, respectively.

## 4. Discussion

In this study we identified a high prevalence of dietary and lifestyle-related risk factors for the health and well-being of women of reproductive age in the eight Latin American countries included. In particular, a high prevalence of being overweight or obese that increases with age, a high prevalence of a sedentary lifestyle, and inadequate intake of essential nutrients were identified, and more than half of the women lived in disadvantaged environments within a low socioeconomic status [31].

Having no access to adequate nutrition and lifestyle conditions compatible with good health violates basic human rights and undermines a women’s ability to achieve well-being and productivity and the health and development of her offspring. In general, women are more vulnerable than men in the Latin American context. According to the Food and Agriculture Organization of the United Nations (FAO) [32] report on food security in Latin America and the Caribbean, obesity affects 27.9% of women <18 years and hence much more than men (20.2%). Accordingly, among the 105 million obese adults in Latin America, 62 million, or 59%, are women. Similarly, data from the ELANS study show a much higher obesity prevalence in women of all age groups (29.5%) than in men (20.6%) [33]. Food insecurity is also more frequently found in women than men, with 69.1 million women living in food insecure conditions (moderate or severe) compared to 54.9 million men [32].

In this study, more than 50% of women of childbearing age had a low socioeconomic status, meaning they were living in poverty, which challenges well-being, as demonstrated, for example, in studies on women of rural communities in south Asian countries [4]. Living in poverty increases the risk for food insecurity and consuming cheap foods with a low density of essential nutrients promotes micronutrient deficiencies (or hidden hunger) [34,35]. This study was conducted in urban settings, which in general showed a similar pattern of loss of quality of life, sedentary habits, limited education, high prevalence of women living in low socioeconomic environments, and poor dietary quality across countries, along with a prevalence of being overweight or obese [36]. Being underweight or obese during pregnancy results in increased maternal morbidity and mortality and represents higher risk for the product of the conception [37], and chances are that women who start pregnancy obese, overweight, or undernourished will remain within these categories, facing all the challenges of surviving pregnancy and making success much more difficult to achieve [38]. In addition, a triple burden of malnutrition with the coexistence of being stunted, overweight, or obese and micronutrient deficiencies is common among children in this region where maternal and child under- and overnutrition coexist [39].

Calcium, iron, and vitamin D are among the essential micronutrients that are important for maternal and child well-being. Iron and calcium deficiencies are associated with increased maternal deaths, and vitamin D has been proposed as a key factor in preserving immunity, particularly to prevent acute respiratory diseases. Thus, in young women adequate intake may contribute to avoiding recurrent illnesses, particularly in vulnerable environments [40].

This study reports frequently inadequate intake of these micronutrients combined with a high prevalence of obesity and being overweight, and higher risks for obesity with adequate intake of calcium and iron. As unequal as the Latin American environment is, we can hypothetically think that some affluent population groups of women with adequate intake of these micronutrients have a trend towards eating better-quality foods, yet live within a setting promoting obesity [41,42]. The ELANS shows short stature in 36% of women, ranging from 23.1% in Brazil to 60.7% in Peru. While genetic factors may contribute to these differences, short height may also result from growth retardation in childhood due to deficient nutrition and recurrent infections [27].

A limitation of this study is that only women in urban areas were considered, thus rural populations were not assessed. The cross-sectional design does not allow causality to be established even with adjustments for covariates. Strengths of the study are the large sample size, the inclusion of nationally representative urban populations of eight countries, the use of two consecutive 24-H recalls, and the inclusion of the analysis of micronutrient intake. Misreporting was included as an adjustment for evaluation of the energy intake, so errors were minimized when evaluating excessive weight.

We wish to emphasize that recent evidence supports a marked impact of maternal nutrition and lifestyle before and during pregnancy on the child’s long-term health, development, and later disease risks, which indicates the need to support adequate living and dietary conditions for women of childbearing age to support the health and opportunities of future generations [43,44]. Public health policies across Latin America should focus more on the health of women of reproductive age to promote achieving the sustainable development goals [45].

## 5. Conclusions

Latin America is a continent of marked inequalities, with enormous gaps between population groups. Disadvantaged environments constitute a vicious cycle for losing human potential of women and their future children. Better understanding of existing challenges for women of childbearing age may contribute to implementing strategies towards breaking the perpetuation of poverty, under- and overnutrition, and non-communicable diseases. Education, promotion of health and healthy lifestyles, and a dietary intake that meets nutrient and micronutrient necessities with the use of supplementation and food fortification may constitute policies to improve equity, opportunities, and the potential for the next generations.

## Figures and Tables

**Figure 1 nutrients-13-00045-f001:**
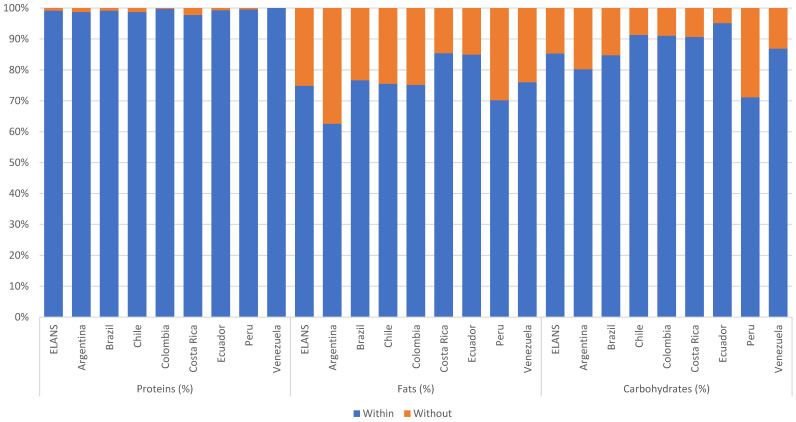
Percentages of individuals in relation to the acceptable percentage ranges of macronutrients among Latin American countries; Latin American Health and Nutrition Study (ELANS), 2015.

**Figure 2 nutrients-13-00045-f002:**
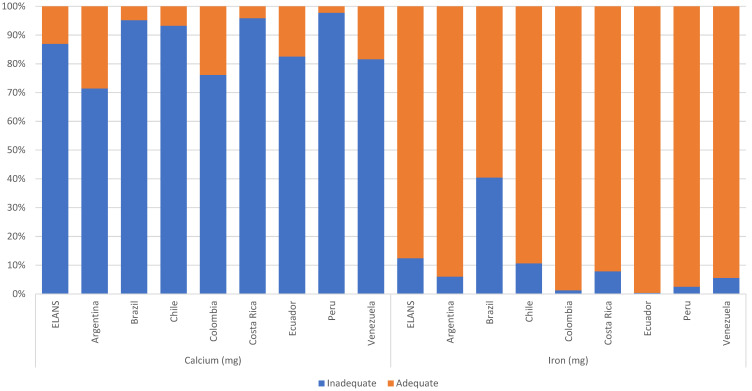
Percentages of individuals by adequacy level of calcium and iron intake among Latin American countries; Latin American Health and Nutrition Study (ELANS), 2015.

**Table 1 nutrients-13-00045-t001:** Sociodemographic characteristics by age group and country among childbearing-age women in the Latin American Study of Nutrition and Health (ELANS), 2015.

	Sample Size	Socioeconomic Level (%)						Educational Level (%)								Marital Status (%)					
Country		High		Medium		Low		None		Basic		Superior—Incomplete		Superior—Complete		Single		Married/Couple		Divorced/Widowed	
		*n*	%	*n*	%	*n*	%	*n*	%	*n*	%	*n*	%	*n*	%	*n*	%	*n*	%	*n*	%
ELANS	3259	286	8.78	1228	37.68	1745	53.54	13	0.40	1.839	56.43	1091	33.48	316	9.70	1431	43.91	1606	49.28	222	6.81
Argentina	465	17	3.66	201	43.23	247	53.12	0	0.00	327	70.32	115	24.73	23	4.95	175	37.63	234	50.32	56	12.04
Brazil	702	52	7.41	321	45.73	329	46.87	10	1.42	286	40.74	347	49.43	59	8.40	316	45.01	341	48.58	45	6.41
Chile	310	28	9.03	149	48.06	133	42.90	0	0.00	190	61.29	85	27.42	35	11.29	158	50.97	137	44.19	15	4.84
Colombia	390	15	3.85	107	27.44	268	68.72	0	0.00	229	58.72	121	31.03	40	10.26	195	50.00	178	45.64	17	4.36
Costa Rica	267	33	12.36	139	52.06	95	35.58	0	0.00	208	77.90	39	14.61	20	7.49	115	43.07	131	49.06	21	7.87
Ecuador	286	35	12.24	102	35.66	149	52.10	0	0.00	237	82.87	34	11.89	15	5.24	105	36.71	161	56.29	20	6.99
Peru	443	83	18.74	143	32.28	217	48.98	0	0.00	101	22.80	300	67.72	42	9.48	166	37.47	254	57.34	23	5.19
Venezuela	396	23	5.81	66	16.67	307	77.53	3	0.76	261	65.91	50	12.63	82	20.71	201	50.76	170	42.93	25	6.31

**Table 2 nutrients-13-00045-t002:** Anthropometric measures by age group and country among childbearing-age women in the Latin American Study of Nutrition and Health (ELANS), 2015.

		15–18 Years						19–34 Years						35–44 Years					
	*n*	*n*	Mean	SD	P25	P50	P75	*n*	Mean	SD	P25	P50	P75	*n*	Mean	SD	P25	P50	P75
**Body Mass Index (BMI)**																			
ELANS	3257	441	23.04	4.50	20.20	22.39	24.80	1899	26.21	5.42	22.39	25.42	29.15	947	28.73	5.87	24.42	28.13	32.00
Argentina	465	44	22.94	3.81	20.67	22.25	24.12	261	25.58	5.40	21.58	24.61	29.06	160	27.90	6.27	23.32	26.30	31.77
Brazil	702	74	22.86	4.94	19.13	22.68	24.93	400	26.13	5.44	22.40	25.37	29.31	228	28.40	5.91	23.63	27.88	32.12
Chile	310	39	23.78	3.57	21.06	23.45	25.33	169	27.33	5.52	23.04	26.32	30.54	102	29.84	6.74	24.88	28.65	33.13
Colombia	390	51	22.09	3.19	20.52	21.64	23.28	240	25.25	5.04	22.02	24.45	27.46	99	26.62	4.79	23.57	26.16	29.06
Costa Rica	267	38	24.43	7.54	20.21	22.72	26.37	160	26.96	5.98	22.85	25.92	29.72	69	31.22	6.55	26.50	30.05	34.65
Ecuador	286	44	23.24	4.13	20.50	22.72	25.46	164	26.27	5.03	22.51	26.08	29.42	78	28.86	4.59	25.50	28.30	30.85
Peru	441	61	23.27	3.59	20.81	23.00	24.77	260	26.44	4.78	22.67	26.25	29.46	120	28.88	5.19	25.65	28.54	30.75
Venezuela	396	60	22.37	4.33	19.76	21.40	24.04	245	26.41	5.99	22.49	25.16	28.88	91	29.86	5.45	26.17	29.43	32.85
**Waist Circumference (cm)**																			
ELANS	3258	441	75.18	10.83	68.00	73.05	80.05	1899	83.62	12.96	74.30	82.00	91.70	947	90.28	13.90	81.00	89.00	98.30
Argentina	465	44	74.32	10.30	68.00	73.05	79.10	261	82.01	13.19	72.80	80.00	89.80	160	88.57	15.14	76.90	87.00	99.10
Brazil	702	74	74.65	13.10	64.00	73.00	81.00	400	83.18	13.40	74.00	81.00	92.10	228	89.21	14.44	79.05	88.00	98.00
Chile	310	39	76.92	8.74	69.40	75.60	82.80	169	86.71	13.93	76.20	84.00	96.00	102	93.17	15.75	80.70	92.85	101.60
Colombia	390	51	73.16	9.29	66.00	71.00	76.60	240	80.09	11.95	72.00	79.00	85.50	99	85.13	12.12	77.60	84.00	93.60
Costa Rica	267	38	81.37	15.51	72.00	77.55	88.00	160	87.11	12.53	78.90	85.00	94.80	69	97.25	13.74	88.00	96.00	105.00
Ecuador	286	44	75.13	9.16	67.50	73.02	80.57	164	84.54	11.23	77.00	84.57	90.55	78	89.42	10.08	82.10	89.09	95.00
Peru	442	61	75.50	8.46	71.00	75.4	79.40	260	84.00	11.42	75.00	84.40	91.65	121	90.70	11.57	83.60	90.60	95.10
Venezuela	396	60	72.88	9.03	65.95	71.00	78.95	245	84.11	14.04	74.20	83.00	92.60	91	93.19	12.80	85.00	92.75	99.15
**Hip Circumference (cm)**																			
ELANS	3257	441	94.31	9.72	88.40	93.20	99.00	1899	100.19	11.13	92.80	99.00	106.25	947	104.57	12.19	96.75	103.00	111.00
Argentina	465	44	95.49	9.05	89.70	94.65	99.70	261	101.57	10.95	93.40	100.00	108.60	160	105.23	12.42	96.35	103.90	111.90
Brazil	702	74	92.69	11.51	85.00	92.75	100.20	400	99.38	12.05	91.00	98.00	106.00	228	103.98	13.44	94.90	102.00	112.00
Chile	310	39	96.70	7.37	91.20	95.20	100.80	169	103.37	12.12	94.50	102.00	109.40	102	105.79	13.13	97.00	103.50	112.50
Colombia	390	51	94.00	8.69	88.80	93.00	97.20	240	98.28	10.40	91.20	97.00	104.00	99	100.21	10.30	94.40	99.50	106.00
Costa Rica	266	38	96.62	14.76	89.00	93.40	102.00	160	102.67	11.61	94.45	101.00	108.40	69	109.63	13.97	100.00	107.40	115.00
Ecuador	286	44	94.48	7.54	89.15	95.50	98.47	164	99.59	9.00	93.00	98.95	104.30	78	103.68	9.25	97.00	102.07	110.00
Peru	442	61	94.00	6.75	90.00	93.20	97.00	260	97.84	8.31	91.20	97.60	102.42	120	102.42	9.49	96.80	101.00	106.40
Venezuela	396	60	92.94	9.93	87.02	90.25	97.77	245	100.95	12.32	94.00	99.55	107.00	91	108.02	11.05	101.60	106.35	112.95
**Neck Circumference (cm)**																			
ELANS	3258	441	31.80	2.76	30.15	31.80	33.10	1899	33.13	3.14	31.00	33.00	35.00	948	34.28	3.39	32.00	34.00	36.10
Argentina	465	44	31.73	2.24	30.50	32.00	33.00	261	32.98	2.67	31.20	33.00	34.50	160	34.07	3.13	32.00	33.90	35.70
Brazil	702	74	30.78	3.82	28.00	30.75	33.00	400	32.21	3.81	30.00	32.00	35.00	228	33.59	4.18	31.00	33.00	36.00
Chile	310	39	32.97	2.47	31.00	32.40	34.60	169	34.62	3.36	32.00	34.00	37.00	102	35.59	3.17	33.80	35.40	37.00
Colombia	390	51	32.08	2.71	30.00	31.60	33.10	240	33.05	2.67	31.30	32.65	34.30	99	33.71	2.72	32.00	33.40	35.20
Costa Rica	267	38	32.38	3.21	30.00	31.30	34.00	160	33.84	2.98	31.55	33.90	35.50	69	35.94	3.18	34.00	36.00	37.50
Ecuador	286	44	31.64	2.60	29.60	32.00	33.00	164	32.82	2.84	31.00	33.00	34.50	78	33.24	2.70	32.00	33.50	34.60
Peru	442	61	31.84	1.94	30.60	31.80	32.80	260	33.10	2.58	31.20	32.50	34.60	121	34.23	2.53	32.70	34.00	35.60
Venezuela	396	60	31.85	1.77	30.62	31.60	33.00	245	33.63	2.98	31.50	33.10	35.10	91	35.27	3.14	33.10	35.20	36.95
**Height (cm)**																			
ELANS	3257	411	158.15	6.56	153.70	158.00	163.00	1899	157.93	6.76	153.00	157.90	162.60	947	157.76	7.04	152.80	157.60	162.10
Argentina	465	44	158.49	6.73	154.10	158.25	163.40	261	159.35	6.49	155.00	159.40	163.30	160	159.04	6.50	155.05	159.00	162.80
Brazil	702	74	161.55	5.92	158.00	162.00	165.00	400	160.44	6.84	155.00	161.00	165.00	228	160.40	7.13	156.00	160.95	165.00
Chile	310	39	158.21	5.81	153.40	157.20	162.60	169	158.25	5.98	154.00	158.80	162.30	102	156.55	6.37	152.10	155.60	159.90
Colombia	390	51	159.95	7.24	155.00	159.00	164.80	240	157.80	6.29	153.20	157.60	161.45	99	158.09	7.95	152.20	158.40	163.20
Costa Rica	267	38	157.04	5.52	152.00	158.05	160.90	160	157.17	6.08	152.65	157.15	161.25	69	156.53	5.38	153.20	156.00	159.60
Ecuador	286	44	155.55	6.38	151.80	157.06	160.10	164	154.81	6.53	150.10	154.02	159.55	78	153.65	6.35	149.40	153.43	157.60
Peru	441	61	154.79	5.04	152.20	155.00	157.50	260	153.30	5.54	149.60	153.30	157.00	120	153.99	5.51	150.00	152.80	157.85
Venezuela	396	60	158.13	6.89	152.53	158.20	163.65	245	159.74	6.31	155.00	160.00	164.10	91	159.34	6.93	154.00	158.50	165.10

**Table 3 nutrients-13-00045-t003:** Anthropometric and physical activity level prevalence by country among childbearing-age women in the Latin American Study of Nutrition and Health (ELANS), 2015.

Country	Nutritional Status (%) *n* = 3057								Waist Circumference (%) *n* = 3258				Neck Circumference (%) *n* = 3258				Height (%) *n* = 3259				Physical Activity Level (%) *n* = 3189					
	Underweight		Normal Weight		Overweight		Obese + Morbidly		Adequate		Inadequate		Adequate		Inadequate		Low		Normal		Low		Moderate		High	
	n	%	n	%	n	%	n	%	n	%	n	%	n	%	n	%	n	%	n	%	n	%	n	%	n	%
ELANS	125	4.09	1137	37.19	1024	33.50	771	25.22	1429	43.86	1829	56.14	2251	69.09	1007	30.91	1172	35.96	2087	64.04	1.949	61.12	897	28.13	343	10.76
Argentina	22	4.73	209	44.95	123	26.45	111	23.87	227	48.82	238	51.18	337	72.47	128	27.53	122	26.24	343	73.76	271	59.04	149	32.46	39	8.50
Brazil	34	4.84	284	40.46	213	30.34	171	24.36	308	43.87	394	56.13	517	73.65	185	26.35	162	23.08	540	76.92	484	70.66	148	21.61	53	7.74
Chile	3	0.97	121	39.03	90	29.03	96	30.97	125	40.32	185	59.68	163	52.58	147	47.42	119	38.39	191	61.61	167	54.75	82	26.89	56	18.36
Colombia	18	4.62	201	51.54	119	30.51	52	13.33	219	56.15	171	43.85	288	73.85	102	26.15	138	35.38	252	64.62	240	62.83	110	28.80	32	8.38
Costa Rica	11	4.12	94	35.21	81	30.34	81	30.34	91	34.08	176	65.92	165	61.80	102	38.20	97	36.33	170	63.67	154	57.89	71	26.69	41	15.41
Ecuador	8	2.80	107	37.41	106	37.06	65	22.73	117	40.91	169	59.09	220	76.92	66	23.08	156	54.55	130	45.45	94	33.94	126	45.49	57	20.58
Peru	8	1.81	164	37.19	171	38.78	98	22.22	178	40.27	264	59.73	311	70.36	131	29.64	269	60.72	174	39.28	259	59.68	136	31.34	39	8.99
Venezuela	21	5.30	157	39.65	121	30.56	97	24.49	164	41.41	232	58.59	250	63.13	146	36.87	109	27.53	287	72.47	280	73.49	75	19.69	26	6.82

**Table 4 nutrients-13-00045-t004:** Energy and macronutrient intake by age group and country among childbearing-age women in the Latin American Study of Nutrition and Health (ELANS), 2015.

		15–18 Years					19–34 Years					35–44 Years				
	*n*	Mean	SD	P25	P50	P75	Mean	SD	P25	P50	P75	Mean	SD	P25	P50	P75
**Energy (kcal)**																
ELANS	3259	1918.88	547.73	1539.71	1859.21	2237.87	1866.23	527.54	1498.60	1817.82	2158.85	1761.74	504.21	1409.53	1695.37	2065.88
Argentina	465	2053.35	582.21	1652.00	1955.00	2491.00	1986.56	569.01	1575.60	1903.96	2270.79	1979.82	539.15	1576.00	1958.93	2321.15
Brazil	702	1765.95	566.94	1352.00	1644.00	2044.00	1703.00	506.33	1310.53	1691.39	1969.69	1654.41	469.84	1295.24	1618.73	1898.62
Chile	310	1633.55	423.70	1334.00	1602.00	1897.00	1549.17	412.81	1238.98	1520.98	1808.00	1463.76	386.16	1224.52	1487.48	1662.39
Colombia	390	2170.30	583.46	1818.00	2141.00	2456.00	2004.75	504.63	1680.80	1981.24	2286.76	2013.97	502.61	1632.82	1962.34	2328.49
Costa Rica	267	1757.53	555.31	1345.00	1694.00	2091.00	1753.77	466.24	1406.41	1695.24	2060.11	1550.32	479.11	1190.10	1536.09	1773.00
Ecuador	286	2041.61	449.22	1810.00	2054.00	2332.00	2142.71	517.14	1791.84	2092.49	2462.64	1900.78	455.63	1612.21	1868.84	2135.00
Peru	443	1979.12	477.27	1590.0	1865.00	2333.00	1989.04	493.20	1668.42	1912.77	2280.53	1850.98	460.04	1476.84	1866.46	2120.50
Venezuela	396	1931.58	529.79	1579.00	1920.00	2124.00	1845.62	481.12	1499.79	1800.91	2085.77	1628.33	406.64	1368.61	1593.77	1897.23
**Protein (% of total kcal)**																
ELANS	3259	15.25	2.80	13.27	15.04	16.92	15.85	2.93	13.94	15.52	17.31	16.17	3.05	14.14	15.73	17.71
Argentina	465	15.63	3.19	13.58	15.30	17.83	15.67	2.44	14.07	15.54	17.16	16.14	2.85	14.44	15.84	17.49
Brazil	702	16.91	3.43	14.42	16.49	19.09	17.41	3.62	15.06	17.04	19.55	17.69	3.56	15.19	17.26	19.58
Chile	310	15.16	2.66	13.30	15.41	16.97	15.87	3.31	14.00	15.37	17.28	16.45	2.76	14.63	16.20	18.03
Colombia	390	14.75	2.41	13.03	14.77	16.21	15.30	2.40	13.76	15.00	16.50	15.13	2.15	13.75	14.96	16.13
Costa Rica	267	14.14	2.30	12.74	14.06	15.28	14.27	2.55	12.49	14.14	15.63	14.43	2.51	12.55	14.62	15.84
Ecuador	286	15.22	2.52	13.70	15.44	16.83	15.71	2.33	14.02	15.53	17.13	15.52	2.21	13.75	15.58	16.99
Peru	443	13.80	1.91	12.11	13.89	15.19	14.81	2.12	13.39	14.73	16.08	14.97	2.49	13.53	14.67	15.96
Venezuela	396	15.58	2.09	14.13	15.67	17.05	16.26	2.45	14.54	16.21	17.67	16.71	2.98	14.39	16.63	18.66
**Fats (% of total kcal)**																
ELANS	3259	30.42	5.68	26.68	30.68	34.22	29.88	5.73	25.97	29.83	33.73	29.93	5.82	25.88	30.09	34.17
Argentina	465	33.21	5.35	30.42	34.13	36.89	33.14	5.29	29.81	33.18	36.67	32.79	5.34	29.69	33.45	36.09
Brazil	702	31.85	4.92	27.66	31.98	34.46	30.72	5.26	27.53	30.30	33.96	31.05	5.28	27.35	30.80	34.84
Chile	310	29.82	4.85	26.68	29.59	32.80	30.30	4.89	26.77	30.11	33.40	30.56	5.79	26.74	30.53	35.04
Colombia	390	32.36	4.83	28.68	32.82	35.65	31.34	5.02	28.08	31.47	34.75	30.30	5.10	26.07	30.72	34.26
Costa Rica	267	30.28	3.76	26.88	30.35	33.13	29.38	5.01	25.66	29.08	32.93	28.55	5.00	25.76	28.17	31.35
Ecuador	286	31.06	3.32	29.30	30.99	32.93	30.54	4.47	27.10	30.74	33.57	30.57	3.89	27.95	30.34	33.53
Peru	443	23.06	3.72	20.35	22.76	25.16	22.89	3.84	20.25	22.76	25.36	22.94	4.23	20.19	22.73	25.43
Venezuela	396	32.45	5.93	28.67	32.15	36.22	30.64	5.10	27.40	30.23	33.92	30.84	5.21	27.26	31.10	34.42
**Carbohydrates (% of total kcal)**																
ELANS	3259	54.33	6.93	49.95	54.05	58.75	54.27	6.84	49.72	54.38	58.84	53.90	7.08	48.72	53.85	58.94
Argentina	465	51.16	7.45	46.72	50.90	55.64	51.19	6.71	46.20	51.09	55.63	51.07	6.91	46.61	50.59	55.06
Brazil	702	51.24	6.13	48.70	51.40	54.62	51.87	6.33	47.92	52.24	56.02	51.26	6.38	46.80	50.58	55.26
Chile	310	55.02	5.86	51.02	55.09	58.75	53.83	6.05	49.87	54.47	57.89	52.99	6.47	47.91	52.92	57.40
Colombia	390	52.90	5.44	48.70	52.88	55.45	53.36	5.76	49.38	53.25	57.07	54.56	5.68	50.80	54.22	58.02
Costa Rica	267	55.58	4.21	53.00	55.84	58.16	56.35	5.89	52.34	56.28	60.62	57.02	6.36	54.14	58.12	60.35
Ecuador	286	53.72	3.94	50.68	53.56	55.71	53.75	5.44	50.22	53.73	57.49	53.90	4.56	49.79	54.06	56.97
Peru	443	63.13	4.70	60.17	63.13	66.26	62.30	4.65	59.57	62.45	65.33	62.09	5.26	59.73	62.17	64.80
Venezuela	396	51.97	6.76	47.90	52.21	55.72	53.11	5.82	49.58	52.98	56.99	52.44	6.26	48.50	52.74	56.42

**Table 5 nutrients-13-00045-t005:** Micronutrient intake by age group and country among childbearing-age women in the Latin American Study of Nutrition and Health (ELANS), 2015.

		15–18 Years					19–34 Years					35–44 Years				
	*n*	Mean	SD	P25	P50	P75	Mean	SD	P25	P50	P75	Mean	SD	P25	P50	P75
**Calcium (mg)**																
ELANS	3259	554.75	235.46	386.85	524.23	693.63	548.52	234.73	379.87	512.24	681.29	531.38	243.54	350.46	490.16	677.39
Argentina	465	689.68	260.59	449.37	687.74	863.98	689.39	221.56	521.09	660.58	826.24	711.18	218.94	533.41	692.28	836.59
Brazil	702	425.49	187.75	282.85	420.32	555.73	420.58	218.69	254.85	389.83	539.84	412.10	216.90	263.03	366.62	518.58
Chile	310	545.16	213.92	391.29	503.77	684.73	475.49	178.99	343.32	443.71	592.24	456.33	224.28	291.17	396.20	559.76
Colombia	390	711.60	253.56	535.24	678.76	806.86	675.26	218.93	514.12	642.96	784.12	707.62	242.98	535.01	675.27	870.42
Costa Rica	267	389.72	145.96	277.09	385.48	520.11	426.69	179.64	293.09	381.04	510.04	388.63	193.61	256.62	320.89	503.96
Ecuador	286	618.59	158.82	487.30	601.75	740.59	658.42	197.10	515.04	629.79	793.87	627.39	172.28	474.40	621.33	726.21
Peru	443	461.47	148.92	368.01	450.93	516.12	449.32	138.89	352.61	427.35	518.43	431.93	152.98	326.77	409.80	510.79
Venezuela	396	640.61	246.94	453.31	596.85	788.79	644.79	237.48	476.13	600.88	793.29	565.76	207.91	386.75	551.33	707.02
**Iron (mg)**																
ELANS	3259	12.90	3.82	10.10	12.53	15.12	12.47	3.79	9.78	12.26	14.86	11.83	3.81	9.11	11.54	14.18
Argentina	465	13.11	3.78	10.15	12.69	15.34	12.95	3.52	10.43	12.57	14.85	13.07	3.25	10.96	13.16	15.00
Brazil	702	10.00	3.31	8.00	9.16	10.82	9.24	3.20	7.05	8.75	10.98	8.71	2.81	6.79	8.34	10.41
Chile	310	11.93	3.40	9.11	11.87	13.48	11.32	2.91	9.30	11.09	12.86	11.20	3.99	9.10	11.03	12.71
Colombia	390	16.40	4.09	13.95	16.31	18.34	15.07	3.54	12.73	15.03	17.26	15.32	3.53	12.74	15.49	17.66
Costa Rica	267	12.60	3.37	10.43	12.34	15.12	12.86	3.37	10.65	12.56	14.65	11.75	2.85	9.49	11.18	13.53
Ecuador	286	13.92	3.33	11.61	14.06	15.99	14.41	3.07	12.16	14.21	16.60	12.83	3.21	10.46	12.20	14.71
Peru	443	13.57	3.24	11.11	13.56	14.63	13.78	3.18	11.34	13.63	15.75	13.36	3.30	11.16	13.24	14.90
Venezuela	396	12.72	2.65	10.66	12.72	14.41	12.57	3.19	10.63	12.25	14.06	11.55	2.84	9.58	11.62	13.60
**Vitamin A (µg)**																
ELANS	3259	558.31	253.92	384.76	520.44	679.50	581.20	291.92	401.75	534.27	693.53	570.05	271.16	398.66	522.26	689.70
Argentina	465	524.54	280.29	346.76	470.60	681.54	529.72	240.89	373.96	499.91	657.66	542.50	212.19	389.61	530.68	651.67
Brazil	702	497.58	283.15	331.61	456.61	578.38	496.91	336.50	309.93	434.45	596.20	516.78	286.48	347.26	465.71	643.63
Chile	310	513.90	127.95	416.32	503.17	589.68	505.32	156.11	385.60	499.66	597.87	483.02	167.93	369.24	467.87	575.43
Colombia	390	693.43	297.71	502.18	627.16	870.92	757.70	341.61	534.07	689.46	907.46	752.13	314.75	533.10	722.33	937.68
Costa Rica	267	532.49	189.38	369.75	499.06	688.98	669.43	310.19	466.85	632.74	821.25	665.87	336.38	435.93	596.83	779.65
Ecuador	286	604.29	299.18	404.78	559.61	720.95	596.76	295.87	429.99	549.18	721.40	593.55	316.84	425.10	525.15	685.52
Peru	443	657.05	215.17	530.44	645.20	752.24	666.12	247.50	505.77	609.18	778.43	647.12	233.54	474.46	627.50	765.59
Venezuela	396	454.27	159.57	335.11	428.59	554.10	494.92	170.43	382.21	481.82	606.74	455.29	145.82	366.24	441.37	558.01
**Vitamin D (µg)**																
ELANS	3259	3.636	2.169	2.058	3.187	4.754	3.494	2.231	1.948	2.946	4.493	3.403	2.250	1.825	2.844	4.442
Argentina	465	3.252	1.436	2.239	3.185	4.071	3.012	1.307	2.040	2.845	3.797	3.141	1.532	2.107	2.838	3.894
Brazil	702	1.999	0.968	1.206	1.913	2.626	1.910	1.096	1.152	1.679	2.419	1.893	0.954	1.151	1.731	2.432
Chile	310	3.823	1.974	2.227	3.545	4.862	2.966	1.569	1.867	2.391	3.787	2.934	1.626	1.829	2.464	3.987
Colombia	390	4.613	1.980	3.278	4.206	5.912	4.612	2.022	3.099	4.167	5.867	4.867	2.265	3.290	4.383	6.051
Costa Rica	267	2.568	1.072	1.839	2.427	3.269	2.720	1.260	1.841	2.501	3.520	2.448	1.427	1.513	2.272	2.909
Ecuador	286	5.722	3.134	3.564	5.265	6.919	6.033	3.361	3.769	5.333	7.454	5.961	3.003	4.066	5.358	6.968
Peru	443	5.022	1.797	3.534	4.654	6.377	5.266	2.161	3.617	4.819	6.418	5.302	2.327	3.649	4.830	6.499
Venezuela	396	2.723	1.376	1.817	2.371	2.942	2.787	1.159	1.974	2.648	3.401	2.570	1.149	1.741	2.206	3.251

**Table 6 nutrients-13-00045-t006:** Association between excess weight and characteristics of childbearing-age women in the Latin American Study of Nutrition and Health (ELANS), 2015.

Variables			Excess Weight			
	Univariate *		Adjusted Model **		Multiple	
	OR	*p*	OR	*p*	OR	*p*
**Age Group**						
15 to 18.9 years	1.00		1.00		1.00	
19 to 24.9 years	2.8875	<0.001	2.6	<0.001	9.87	<0.001
35 to 44.9 years	6.487	<0.001	5.75097	<0.001	19.80	<0.001
**Educational Level**						
None	1.00		1.00			
Basic	1.117	0.843	1.8597	0.279	---	---
Superior—Incomplete	0.982	0.974	1.65055	0.384	---	---
Superior—Complete	0.949	0.926	1.6077	0.418	---	---
**Physical Activity**						
Low	1.00		1.00			
Moderate	1.116	0.0176	1.12	0.191	---	---
High	0.99	0.914	0.90	0.362	---	---
**Neck Circumference**						
Low	1.00		1.00		1.00	
High	5.674	<0.001	5.46	<0.001	11.99	<0.001
**Country**						
Argentina	1.00		1.00		1.00	
Brazil	1.192	0.142	1.22	0.102	1.50	0.006
Chile	1.481	0.008	1.51	0.007	1.15	0.427
Peru	1.544	0.001	1.63	<0.001	1.91	<0.001
Colombia	0.771	0.059	0.91	0.485	0.93	0.649
Costa Rica	1.523	0.007	1.37	0.05	1.50	0.029
Ecuador	1.468	0.012	1.57	0.004	1.98	<0.001
Venezuela	1.21	0.166	1.23	0.085	1.30	0.105
**Calcium Intake**						
Inadequate	1.00		1.00		---	---
Adequate	0.720	0.002	0.95	0.623	---	---
**Vitamin D Intake**						
Inadequate	1.00		1,00		1.00	
Adequate	1.863	0.046	2.41	0.006	2.12	0.026
**Iron Intake**						
Inadequate	1,00		1.00		1.00	
Adequate	0.873	0.208	1.28	1.407976	1.34	0.02
Total fat (g)	0.9887611	<0.001	1.00	0.060	---	---
Total protein (g)	0.9905252	<0.001	1.00	0.64	---	---
Total carbohydrates (g)	0.9973948	<0.001	1.00	0.278	---	---

* Unadjusted model—univariate model. ** Adjusted for misreporting and social economic level. OR = Odds ratio

## Data Availability

The data presented in this study are available upon request from the corresponding author. The data are not publicly available due to privacy or ethical restrictions.

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
