# Peer review of "Childbearing Age Women Characteristics in Latin America. Building Evidence Bases for Early Prevention. Results from the ELANS Study"

_nutrients, 2020, doi:10.3390/nu13010045_

Round 1

Reviewer 1 Report

The manuscript discusses in detail the population data for female population in Latin American countries, nutrient levels and associated disease conditions. The study design selected by the authors is very thorough and test methods used for data collections are scientifically validated. I think the sample size in the study was large enough to generate statistically significant data and conclusion. The relationship of socioeconomic status of these countries with macro/micro nutrients levels has been discussed and I think, it's a critical factor which the authors have described and discussed fully.

Very well written manuscript studying a major reality for the eight countries described. There are few spell and grammar checks that needs to be corrected. 

For example,

Latin America is a continent of marked inequities  inequalities.

Author Response

In line 79 the word inequities was substituted by inequalities as well as in line 312 

In line 205 the word greatest was substituted by highest 

in line 259 the work within was included 

all those highlighted in yellow 

Reviewer 2 Report

This well written manuscript addresses the lifestyle and dietary characteristics of women of childbearing age, through sociodemographic, dietary intakes of energy, proteins and micronutrients, physical activity and anthropometric characteristics.

The study data is from a very large cohort of The Latin American Study in Nutrition and Health/Estudio Latinoamericano de Nutrición y Salud  (ELANS) which is a household-based, multicentric cross-sectional survey,  including 3259  women aged between 15 and 45 years. from urban populations in  eight countries (Argentina, Brazil, Chile, Colombia, Costa Rica, Ecuador, Peru, and Venezuela). The fieldwork for the ELANS study was conducted from 2014 to 2015.  

Introduction section –

Lines84-86 please add the references Tiffon, C. The Impact of Nutrition and Environmental Epigenetics on Human Health and Disease. Int. J. Mol. Sci. 201819, 3425.

Results section –

Table 3 nutritional status N of women is 3057:::::

Lines 206-211 -Most of the countries showed mean protein intakes matching recommended intakes. Argentina, Peru,  Colombia, Chile, and Venezuela had more than 20 % of their population with a fat intake higher than  recommended intake range, except for Peru which was the only country with mean fat intake below  recommendations. -  Please be more clear.

Line 245- Women 19 years and older had a 19.8-fold higher risk for excess weight  than younger women-according the  table 6 Women 35- 44.9 9 years old had a 19.8-fold higher risk for excess weight  than younger women

Author Response

Response to Reviewer #2

Thank you for your comments!

Point 1: Introduction section –

Lines84-86 please add the references Tiffon, C. The Impact of Nutrition and Environmental Epigenetics on Human Health and Disease. Int. J. Mol. Sci. 201819, 3425.

It was added as reference # 17 and a additional comment in lines 84-89 included.

Point 2: Table 3 nutritional status N of women is 3057:::::

Yes, we included totals on nutritional status and other variables: waist and neck circumferences, stature and Physical activity level

Point 3: Lines 206-211 -Most of the countries showed mean protein intakes matching recommended intakes. Argentina, Peru,  Colombia, Chile, and Venezuela had more than 20 % of their population with a fat intake higher than  recommended intake range, except for Peru which was the only country with mean fat intake below  recommendations. -  Please be more clear.

We explained that while Argentina, Peru, Colombia, Chile and Venezuela had more than 20% of their female population with a fat intake higher than recommended intake range, Peru needs to be highlighted as the mean intake was lower, indicating that mean values need to be cautiously analyzed, as some values might be at extremes, thus making the average not fully demonstrative of the majority

Point 4: Line 245- Women 19 years and older had a 19.8-fold higher risk for excess weight  than younger women-according the  table 6 Women 35- 44.9 9 years old had a 19.8-fold higher risk for excess weight  than younger women

In line 245 now line 249 we explained better according to group ages :  Women 19 to 24.9 years had a 9.87-fold higher risk for excess weight and 35-44.9 years old had a 19.80 -fold higher risk for excess weight than female adolescents.